# DNA Damage Response−Related Proteins Are Prognostic for Outcome in Both Adult and Pediatric Acute Myelogenous Leukemia Patients: Samples from Adults and from Children Enrolled in a Children’s Oncology Group Study

**DOI:** 10.3390/ijms24065898

**Published:** 2023-03-20

**Authors:** Stefan E. Hubner, Eduardo S. de Camargo Magalhães, Fieke W. Hoff, Brandon D. Brown, Yihua Qiu, Terzah M. Horton, Steven M. Kornblau

**Affiliations:** 1Department of Leukemia, The University of Texas M.D. Anderson Cancer Center, Houston, TX 77030, USA; 2Department of Ageing Biology/ERIBA, University Medical Center Groningen, 9713 AV Groningen, The Netherlands; 3Department of Internal Medicine, UT Southwestern Medical Center, Dallas, TX 75390, USA; 4Division of Pediatrics, The University of Texas M.D. Anderson Cancer Center, Houston, TX 77030, USA; 5Department of Pediatrics, Dan Duncan Cancer Center, Texas Children’s Hospital, Houston, TX 77584, USA

**Keywords:** AML, proteomics, RPPA, DNA damage

## Abstract

The survival of malignant leukemic cells is dependent on DNA damage repair (DDR) signaling. Reverse Phase Protein Array (RPPA) data sets were assembled using diagnostic samples from 810 adult and 500 pediatric acute myelogenous leukemia (AML) patients and were probed with 412 and 296 strictly validated antibodies, respectively, including those detecting the expression of proteins directly involved in DDR. Unbiased hierarchical clustering identified strong recurrent DDR protein expression patterns in both adult and pediatric AML. Globally, DDR expression was associated with gene mutational statuses and was prognostic for outcomes including overall survival (OS), relapse rate, and remission duration (RD). In adult patients, seven DDR proteins were individually prognostic for either RD or OS. When DDR proteins were analyzed together with DDR−related proteins operating in diverse cellular signaling pathways, these expanded groupings were also highly prognostic for OS. Analysis of patients treated with either conventional chemotherapy or venetoclax combined with a hypomethylating agent revealed protein clusters that differentially predicted favorable from unfavorable prognoses within each therapy cohort. Collectively, this investigation provides insight into variable DDR pathway activation in AML and may help direct future individualized DDR−targeted therapies in AML patients.

## 1. Introduction

Acute myelogenous leukemia (AML) is a heterogeneous cancer of the blood and bone marrow characterized by numerous genetic and epigenetic changes that culminate in different patterns of protein expression. AML begins in the bone marrow of certain bones and often quickly transitions into the blood, but the cancer can also spread to other parts of the body including the liver, the spleen, lymph nodes, and the central nervous system. AML represents roughly 15–20% of acute leukemia cases in children and about 80% in adults [1]. The standard of care currently combines results of cytogenetics (a diagnostic procedure that detects chromosomal aberrations) with targeted testing for mutations in genes such as FLT3, NPM1, and CEBPA to classify prognostic groups [2]. Relapse remains common in AML and occurs in approximately 45% of the younger and the majority of elderly patients [3]. Consequently, efforts to better understand cellular resistance mechanisms in leukemia may help to dramatically improve the survival of patients affected by this deadly disease.

DNA damage response (DDR) is essential for preserving genomic integrity and active mitotic control and is therefore crucial to the survival of malignant leukemic cells. DDR genes are frequent mutational targets in many cancers, and alterations in DDR activity have been shown to contribute to carcinogenesis as well as response to therapy and prognosis [4,5]. Like many other cancers, mutations in DDR genes are seen in AML, though less commonly. However, many studies have shown that DDR function is often abnormal in AML [6,7,8]. DDR abnormalities including loss of function as well as upregulation of activity are thought to contribute to both leukemogenesis as well as modulating therapy response and chemotherapy resistance, which is associated with poor treatment outcome and unfavorable prognosis in AML patients [9]. If DDR activity is infrequently mutated, but commonly abnormal, this suggests that there are likely alterations at the protein level, resulting from the combined consequence of mutations, epigenetic modulations, and responses to internal and external signaling, which integrate to cause the altered DDR function seen in AML cells. We hypothesized that there would be a finite number of recurrent DDR protein expression and post−translational modification (PTM) patterns in AML cells, and that these would correlate with outcome, and which might also suggest how to personalize therapies using DDR−targeting agents. While previous investigations have only minimally characterized the activity of DDR protein expression in AML, our study provides detailed insight into the activation patterns of DDR proteins as well as novel DDR−related protein groupings.

This study used Reverse Phase Protein Array (RPPA), an antibody detection−based technology, to quantify the relative levels of protein expression in a cohort of 810 adult AML patients and in a separate cohort of 500 pediatric patients [10]. RPPA has the distinct advantage over alternative methodologies for quantifying protein expression levels as it simultaneously assesses hundreds of total and post−translationally modified (PTM) proteins in many samples. Additionally, RPPA requires reduced patient sample volume as compared to other techniques such as ELISA or mass spectrometry. For these reasons, and in consideration of the inevitable convergence of all genetic and environmental influences on protein expression, RPPA and proteomic analysis provides meaningful insight into the underlying biology of AML.

We have validated over 550 antibodies for use on RPPA, across a broad range of functions and systems within the cell. This includes 21 proteins known to directly participate in DDR which constitute the primary DDR functional group in our analysis. These proteins include DDR sensors, mediators, transducers, effectors, and targets involved in both single and double strand break recognition and repair as shown in Figure 1 (Hoff, et. al. *Int. J. Mol. Sci.*
**2023**, *24*, 5460.) of our accompanying manuscript submitted in this Special Issue. This analysis also includes several non−canonical DDR proteins that demonstrated significant correlation with the core DDR proteins, including PDCD1, whose expression is modulated by DNA double−strand break repair pathway activation [11]. PDCD1 is also known to be affected by alternative DDR pathways including DNA damage ATR/Chk1 checkpoint signaling [12,13,14].

For subsequent analyses, expanded sets of DDR−related proteins containing cell cycle checkpoint proteins were used. CDK protein expression directly impacts DDR, as the induction of cell cycle arrest is necessary for DNA repair and CDK levels control cell cycle progression. Previous reports have documented direct modulations between DNA repair and, for instance, CDK9 [15], CDKN1B [16,17,18], CDKN1B.pS10 [19], CDKN1B.pT198 [20,21], CDKN2A [22,23], CDC25C [24,25], CDK1_2_3pT14 [26,27], and AURORA_A_B_C [28]. In this report, we reveal hitherto unique patterns of DDR protein expression in AML and characterize their association with distinct clinical outcomes.

## 2. Results

### 2.1. DDR Protein Expression and Activation Status Are Abnormal in Adult AML

The levels of protein expression of 21 DDR proteins were measured in 810 newly diagnosed adult AML patient samples, and unbiased hierarchical clustering identified four optimal clusters (C1−C4), as shown in Figure 1. Most cases displayed expression of the individual DDR proteins that were different from that of the normal CD34+ control cells (Table 1: Adult Patients). In adult AML there were four proteins with markedly low expression across nearly all (>90%) cases (deep blue on the color scale: CHEK1, MSH2, MSH6, and XPF), another six with expression below >50% of cases, and four with downregulated expression across >25% of cases. Four proteins had upregulated expression in over 25% of cases (warm colors on the color scale) including the activated PTM forms of CHEK1.pS345 and CHEK2.pT68, and the total forms of DDB1 and XPA. Overall, the level of expression was abnormal in 18 of 21 proteins, in >75% of cases for seven proteins, in >50% for three proteins, and in >25% of cases for another eight proteins. Thus, nearly half of the DDR proteins assayed have protein expression levels that are abnormal in most AML cases. These dysregulations at the protein level align with findings from prior gene expression profiling data detailing abnormalities in DDR in leukemia. Furthermore, since all four PTM (activated) forms were abnormal in most cases, this suggests that DDR activation and activity is also dysregulated in most cases of AML.

### 2.2. DDR Protein Expression Correlates with Clinical Characteristics, and Is Prognostic for Remission Duration (RD) in Adult AML

The distribution of clinical characteristics, cytogenetic and mutational events, and outcome measures for the entire population and within each cluster is shown in Table 2. Clinical variables proportionally associated with all clusters included gender, race, performance status, cytogenetics risk group, and prior malignancy, among others. Bone marrow (BM) blast percentage displayed significant variance across clusters where C4 displayed the lowest BM blast percentage (*p* < 0.001). Additionally, white blood cell (WBC) count (*p* < 0.001), hyperleukocytosis (*p* = 0.006), peripheral blood (PB) blast percentage (*p* < 0.001), PB absolute blast count (*p* < 0.001), and platelet count (*p* = 0.024) were all significantly associated with the clusters. Mutations in FLT3-ITD (*p* < 0.001), IDH1/2 (*p* = 0.003), and NPM1 (*p* < 0.001) also significantly varied, with a concentration of these events in C1 patients. Notably, C4 contained the highest proportions of patients displaying RUNX1 and TP53 mutations (*p* = 0.047 and *p* = 0.008, respectively).

The adult AML DDR cluster membership was highly prognostic for multiple outcome measures. Although the rate of complete remission (CR) attainment did not differ (*p* = 0.411), DDR protein cluster membership was highly predictive of outcome after attaining CR, as both the relapse rate and RD were very different across the clusters. Only 25% of C4 and 33% of C3 patients relapsed, vs. 50% in C1 and 54% in C2 (*p* = 0.011). Similarly, the percentage of patients in remission at 5 years was highest in C4 (74%) and C3 (60%) and declined to 44% in C1 and 40% in C2 (*p* = 0.009). The small number of cases in C4 likely precludes statistical significance in some comparisons, despite the apparent visual differences. the comparison between C4 (prognostically best for RD) and C1 and C2 (prognostically worst for RD) is interesting. C4 was characterized by significantly increased average levels XPA, CHEK2.pT68, and RAD51 and lower levels of total CHEK2, especially compared to C1, while BABAM1.pS29 was higher in C1 but not in C2 (Figure 1A).

### 2.3. A Subgroup of DDR Proteins Is Prognostic for Overall Survival (OS)

A subset of the DDR proteins including BABAM1.pS29, BAP1, MSH2, MSH6, RAD50, RAD51, RPA32, RPA32.pS4_8, SSBP2, TP53BP1, VCP, and XPF, was found to be prognostic for OS with the patients divided into six clusters (*p* = 0.0023) (Figure 2A,B). At 4 years, cumulative OS probability for the poorest performing C6 diminished to only 10%. In contrast, three of the remaining clusters (C1, C3, C5) maintained 30–35% OS probability even at six years. Notably, when comparing the prognostically similar C1 (red), C3 (yellow), and C5 (pink) vs. the prognostically adverse C2 (blue), C4 (lime), and the very adverse C6 (green), the favorable clusters had higher RPA32, RAD50, BABAM1.pS29, and BAP1, with C1 and C3 also having higher SSBP2, MSH6, and TB53BP1. RAD51 was higher in the three adverse clusters compared to the three favorable clusters, with VCP being highest in C6.

### 2.4. Seven DDR Proteins Are Individually Prognostic for Either RD or OS

Next, we examined whether each individual protein from the set of 21 DDR proteins stratified the outcomes of OS and RD by predefined divisions of expression, namely medians, terciles, and sextiles. Among the 21 proteins, 7 were found to be individually prognostic: BABAM1.pS29, DDB1, MSH6, RAD50, SIRT6, TP53BP1, and XPF. For RD, abnormal levels of DDB1, either higher or lower than normal, and higher levels of SIRT6 were prognostically adverse (*p* = 0.01 and *p* = 0.021, respectively) (Figure 3A,B). For OS, lower levels were adverse for BABM1.pS29 (*p* = 0.0006), MSH6 (*p* = 0.0025), RAD50 (*p* = 0.002), TP53BP1 (*p* = 0.01), and XPF/ERCC4 (*p* = 0.006) (Figure 3C–G).

### 2.5. Novel DDR and DDR-Related Protein Groupings Are Prognostic for OS

In recognition of the emerging crosstalk that occurs between DDR proteins and other DDR-related proteins operating in diverse cellular signaling pathways [10], we constructed a new grouping of proteins based on both DDR and DDR-related proteins (Expanded-DDR set = ExpDDR) (Figure 4A). ExpDDR had six clusters of patients that displayed different OS patterns (Figure 4B) with C1 (red 15.5% of cases, *n* = 126) performing the best, and C6 (10.8% of cases, *n* = 88) doing significantly worse than four of the five other clusters. Multivariate analysis by Cox regression demonstrated cluster membership to be independently prognostic for OS along with age, secondary AML, complex karyotype, and NPM1 and TP53 mutations (Appendix A). We searched for a subset of ExpDDR that was maximally prognostic of survival and identified a grouping of eight that included four previously recognized DDR proteins: TP53BP1, XPF, RAD50, BABAM1.pS29, and four of the DDR-related proteins: CDK1_2_3.pT14, TP53, CDK9, and AURKA (Figure 4C,D). The poorest performing C4 demonstrated elevated relative levels of AURKA and CDK1_2_3.pT14. This pattern was modestly conserved in the poor performing C6 of the previous DDR grouping (as shown in Figure 4A,B). Notably, RAD50 and especially TP53BP1 expression was higher in the best prognosis C1 relative to the other clusters (Figure 4C,D), similar to C1 and C3 in Figure 2.

### 2.6. DDR-Related Proteins Are Prognostic for OS in Patients Receiving Either Conventional Chemotherapy (CC) or Venetoclax plus Hypomethylating Agent (HMA) Therapy (VH)

Recently, the choice of AML therapy in adults has divided into a decision between CC (most commonly cytosine arabinoside and an anthracycline, often combined with other agents), or therapy with a BCL-2 inhibitor (typically venetoclax) combined with a HMA (decitabine, azacytidine) (VH). As these therapeutic strategies have different mechanisms of action, we hypothesized that DDR and DDR-related proteins might be differentially prognostic of response to CC or VH.

We first asked if individual proteins from the broader set of DDR and DDR-related proteins stratified OS in CC/VH patients by predefined divisions of expression: medians, thirds, quartiles, quintiles, and sextiles. This protein group contained those delineated in Figure 4A (ExpDDR) with the addition of TP53, CDKN1A, CDKN1B, CDKN1B.pS10, CDKN1B.pT198, and CDKN2A, to constitute a maximally expanded array of DDR and DDR-related proteins [10]. Appendix A contain the *p*-values for each prognostic protein for every division analyzed, in the CC and VH patients, respectively. When considering only CC patients (*n* = 340), XPF, TP53BP1, BABAM1.pS29, and RPA32 were prognostic for OS at all quantiles analyzed (Appendix A). In contrast, for VH patients (*n* = 79), CDKN1B and CDKN1B.pS10 were highly prognostic for OS analyzed at all quantiles (Appendix A).

We next determined if this comprehensive collection of DDR-related proteins was globally prognostic for OS in CC patients and VH patients. Unbiased hierarchal clustering stratified CC patients into four clusters with significantly different OS (Figure 5A,B). Relative to the best performing C1, the poorer performing clusters displayed downregulated TP53BP1, MSH6, and SSBP2. Notably, CDK1_2_3.pT14 was upregulated in the poorest performing C4 relative to the other clusters. Multivariate analysis demonstrated multiple CC clusters to be independently prognostic for OS along with age, secondary AML, complex karyotype, −5/5q-, t(8;21), and inv(16) (Appendix A). In VH patients, three DDR-related proteins, CDK9, CDKN1B, and CDKN1B.pS10, stratified patients into three clusters whereby C1 demonstrated significantly improved OS relative to the other clusters (Figure 5D,E). Relative to the poorly performing C2 and C3, C1 was marked by lower levels of CDKN1B and CDKN1B.pS10. Both univariate and multivariate analysis showed VH cluster membership to be independently prognostic for OS in addition to IDH and PTPN11 mutations (Appendix A).

Unsurprisingly, there were strong associations between the six ExpDDR clusters defined in Figure 4A and the responsiveness to CC versus VH therapy, as shown in the ‘Treatment Subcluster’ annotation at the top of the heatmap (Figure 4A). Notably, patients in the ExpDDR C1 (red) who received CC were almost exclusively (*n* = 68/69) found in the most favorable prognosis CC C1, while most patients from the ExpDDR C1 treated with VH (*n* = 11/13) were clustered in the very-poor prognosis VH C2 and VH C3, highlighting that patients with the ExpDDR C1 protein expression profile do better with CC (Figure 5C,F). In contrast, cases in ExpDDR C2 (blue) treated with CC were predominantly in the poor performing CC C3 (*n* = 39/43), while more than half of those treated with VH (*n* = 9/16) were in the more favorable VH C1, suggesting that the ExpDDR C2 protein expression pattern was associated with more resistance to CC than to VH. The responses in other ExpDDR clusters were independent of DDR effect: ExpDDR C4 (violet) was more common in the favorable CC C1 and C2 (*n* = 63/71) and favorable VH C1 (*n* = 10/18), but was present in all clusters, while ExpDDR C3 (lime green) was relatively evenly split between favorable CC C2 and VH C1 (42 cases) and unfavorable CC C3 and C4 and VH C2 (28 cases). We therefore observe that the protein expression patterns of the ExpDDR protein set demonstrate different prognostic consequences depending on the therapy used.

### 2.7. DDR Protein Expression and Activation Status Are Abnormal in Pediatric AML

The level of protein expression of a separate set of 21 DDR proteins were measured in 500 newly diagnosed pediatric AML patients and, analogous to the adult cases, most pediatric cases displayed expression of the individual DDR proteins that was different from that of the normal CD34+ control cells (Table 1: Pediatric Patients). In pediatric AML, four proteins were downregulated in expression in over 50% of cases while six proteins were upregulated in over 25% of cases. In total, expression was abnormal in 17/21 proteins, and in all the activated PTM proteins that were analyzed. CHEK1, MSH2, and SSBP2 were markedly low in >75% of cases in both pediatric and adult patients while the activated PTM forms of CHEK1.pS345 and CHEK2.pT68 were upregulated in >25% of cases across each patient cohort. When the proportion of proteins that had abnormal expression was compared between pediatric and adult patients, regardless of whether the cut-points from Table 1 were used (chi-square = 0.60, *p* = 0.6), or more simply the percentage of cases with low, equal, or high expression (chi-square = 1.73, *p* = 0.42), or the percent of abnormal proteins (>25%, >50%, or >75%) (chi-square = 0.36, *p* = 0.83), the distributions were very similar across the two age cohorts. Thus, compared to adults, pediatric patients displayed a similar distribution of dysregulated DDR protein activity and activation.

### 2.8. DDR Protein Expression Reveals Five Pediatric Clusters with Significantly Different Prognoses

Unbiased hierarchical clustering identified five optimal pediatric clusters displaying unique clinical characteristics, as shown in Figure 6. The distribution of clinical characteristics, cytogenetic and mutational events, and outcome measures for this population was previously described by Hoff et al. [10]. Notable upregulated proteins in the unfavorable prognostic groups (C1-C3) included SSBP2, MSH2, MSH6, RPA32.pS4_8, SIRT1, RAD50, XRCC1, RPA32, CHEK2, and ERCC1 (Figure 6 A). Conversely, BRCA2 was upregulated in the remaining favorable prognostic clusters, especially C5, compared to those patients with poorer prognosis. Event-free survival (EFS) (*p* = 0.025) and remission duration (*p* = 0.058) also varied significantly between the five clusters (Figure 6B,C). Compared to the combination of the three poor prognosis clusters (C1, C2, C3), the two favorable clusters (C4, C5) had highly significantly superior OS (*p* = 0.023), RD (*p* = 0.0049), and EFS (*p* = 0.0037) (Figure 1).

## 3. Discussion

In this study, we confirmed our hypothesis that a finite number of recurrent patterns of DDR protein expression patterns are observed in AML, and that these are associated with clinical and molecular features, as well as with several outcome measures. In both adults and pediatric patients, global DDR protein expression was highly prognostic for relapse, RD, and OS. Significant associations were also found between clusters and clinical features including WBC count, % BM blast, and TP53 mutational status. In adults, unbiased hierarchal clustering identified a subgroup of DDR proteins and an expanded set of DDR and DDR-related proteins (ExpDDR), each of which were prognostic. Multivariate analysis also confirmed that ExpDDR, CC, and VH clusters were all independently prognostic for OS. When CC and VH clusters from Figure 5 were mapped onto the ExpDDR clusters from Figure 4A, dramatic differences in prognosis between CC and VH patients were seen in ExpDDR C1 and C2. Patients in ExpDDR C1 who received CC almost always belonged to the most favorably surviving CC C1, while the majority of those classified as ExpDDR C2 who received VH demonstrated the best survival (VH C1) among the VH clusters. Conversely, ExpDDR C1 patients treated with VH and ExpDDR C2 patients treated with CC consistently fell into poor prognostic clusters. Taken together, ExpDDR C1 may be used to identify patients who respond well to CC but not to VH, while ExpDDR C2 membership may predict increased therapeutic resistance to CC rather than to VH.

In adults, we observed that the expression levels of most studied proteins were abnormal, with many being low in over 50% of patients (*n* = 10/21) or higher than normal in >25% of cases (*n* = 4/21), and with all but three proteins having abnormal levels in >25% of cases. Highly comparable results were seen in the pediatric cohort. These findings highlight that DDR protein expression is abnormal in AML, regardless of age group, in corroboration with prior mRNA-based gene expression profiling findings of high DDR protein expression in AML [30,31]. Furthermore, as all four of the PTM forms corresponding to activated states were frequently abnormal, this demonstrates that DDR protein activation is also abnormally dysregulated in AML. While DDR protein activation was not discernable from the mRNA GEP studies, several other reports note connections between leukemia and abnormalities in activated DDR proteins such as CHEK2.pT68 and CHEK1.pS345 [32,33]. Our data suggest proteins for therapeutic modulation across all AML cases, and other proteins for modulation in selected cases, to test if impairing DDR activity can increase sensitivity to existing therapies. Examining how expression levels fluctuate between diagnosis and relapse may also suggest which DDR proteins to target to re-sensitize resistant or relapsed disease. Below, we highlight some specific considerations.

We revisualized our findings by analyzing the median expression of each protein within a given cluster, with respect to its functional role in DDR as per Esposito et al. [34] to provide a more holistic view of how DDR is globally being affected (Figure 7). Using the cluster assignments from the first model, as shown in Figure 1A, we observed that C4, the most prognostically favorable cluster, is characterized by lower levels of expression across diverse DDR functional areas including sensors, mediators, and both double and single-stranded repair mechanisms, with the exception of high levels of XPA. This may reflect the reduced capability of these cells to recognize and repair DNA damage, and this may be a component of their increased sensitivity to leukemia chemotherapy, which relies on the production of DNA damage. Notably, C2 (blue, worst prognosis) is characterized by increased activity across multiple DDR functional areas including BER and BRCA function compared to the more favorable C4 and C3. This may be indicative of a generalized increased capacity to recognize and repair chemotherapy-induced DNA damage, thereby promoting inferior prognoses. C1 (red) displays a poor prognosis in line with C2, but C1 has increased activity in differing DDR functional areas including sensors, mediators, HE/NHEJ, and BRCA. Notably, XPA activity is high in favorable clusters and low in unfavorable clusters, suggesting that therapeutic inhibition of its activity would be detrimental. Moreover, levels of XRCC1 and BABAM1.pS29 are higher in the less favorable clusters, suggesting these are targets for inhibition, either singly, or potentially in combination.

Bullinger et al. showed that Core Binding Factor (CBF), AML t(8;21), and inv(16) that displayed higher mRNA levels of CHEK2 and RAD51, had a more unfavorable outcome [30]. As shown by the individual box plots (Appendix A), we verified that CHEK2 expression was marginally higher in both t(8;21) and inv(16) AML while RAD51 was lower in inv(16) and equal in t(8;21) compared to non t(8;21) cases. We found higher CHEK2 protein levels in the favorable cytogenetics group and that neither CHEK2 nor RAD51 were individually significantly prognostic for OS or RD. However, Bullinger’s connection between elevated RAD51 mRNA levels and unfavorable outcomes, combined with our findings of RAD51 upregulation in adverse prognostic groups (Figure 2B), provide impetus to examine CYT01B and other RAD51 inhibitors as targeted therapy options for these patients [35,36]. 

In AML with a complex aberrant karyotype, Schoch and colleagues found upregulation of MSH6 mRNA [31], while our study found that increased levels of MSH6 were associated with a more favorable OS. We also found the highest MSH6 expression in the most favorably surviving C1 of Figure 4A, which simultaneously displayed minimal complex karyotype signatures. Though our findings differ from previous reports, it is important to recognize that mRNA expression may not directly correlate with the translated protein expression. MSH6 protein downregulation may signal reduced cellular capacity to synthesize DNA repair enzymes that would otherwise protect from the accumulation of cancer-associated genetic defects. The aggregation of cellular mutations either in the nucleus or in the mitochondria will likely precipitate oxidative phosphorylation damage and the ensuing metabolic shift towards increased malignant fermentation [37].

Two additional studies have shown activated CHEK1 in complex karyotype AML [38,39], aligning with our finding in Figure 4A that while CHEK1.pS345 is high in many patients, expression is slightly more elevated in the complex karyotype sub-groups of C3 and especially C6. Additionally, while these previous studies noted low RAD50 in complex karyotype AML, we too found lower RAD50 protein expression in complex karyotype patients (Appendix A). 

Previous studies have characterized an emerging role of CDK cell-cycle-independent functions in DDR [15,16,17,18,19,20,21,22,23,26,27,40,41]. CDK1_2_3.pT14 upregulation in the poorest surviving CC patient clusters of the DDR-related protein networks (Figure 5A) may signal enhanced cell cycle regulation and increased opportunity to perform DDR, thus highlighting the relevance of CDK inhibition in AML. A study published in 2022 showed that CDK7 inhibitors suppressed both leukemia stem cell (LSC)-enriched subsets in vivo and synergized with the BCL-2 inhibitor venetoclax [42]. Further studies are needed to examine potential CDKi synergy with chemotherapeutic agents such as Cytarabine and Doxorubicin, which may help to improve therapy response among select CC patients.

BRCA2 downregulation may reflect reduced tumor suppressor function in the unfavorable pediatric AML clusters (Figure 6). Indeed, studies have shown that cancer cell proliferation is enhanced by the downregulation of BRCA2 expression [43]. Low levels of BRCA2 in poor prognosis pediatric AML clusters also aligns with findings of increased risk of leukemia in patients with BRCA2 mutations [44]. Mutated or downregulated BRCA2 would lead to reduced associations with Rad51 and subsequent inhibition of homologous recombination repair of DNA double-strand breaks. Impairment of DNA repair functions would lead to the accumulation of genetic damage and may favor disease malignancy.

A number of therapeutic agents are undergoing Phase I trials for proteins affected by DDR mutations and may therefore serve as potential targets in AML patients with abnormalities in DDR protein expression (Table 3). Notably, we identified several proteins that displayed higher levels of expression in prognostically adverse cohorts. As none of these molecules appear in Table 3, our study of DDR proteins from patient-derived samples has identified several clinically relevant targets for new agent development. Since higher expression correlated with improved outcome for many of the identified individually prognostic DDR proteins, strategies designed to elevate, for example, MSH6 and TP53BP1, are also merited. Several protein inhibitors may be appropriate when considering therapy options for either CC or VH patients. For our CC patients, higher CDK1_2_3.pT14 levels associated with poorer OS and thus the efficacy of CDK inhibitors such as THZ531, THZ1, YKL-5-124, and LDC4297, may likely merit further investigation in the context of chemotherapy-treated AML patients [42]. In VH patients, low levels of CDKN1B and CDKN1B.pS10 differentiated two poor surviving clusters from a favorable cluster (Figure 5D,E). Hence, therapeutic methods involving the induction of these proteins may help restore sensitivity to VH in these patients. Conversely, the poorest surviving VH cluster displayed marked relative upregulation of CDK9 (Figure 5D,E). Several CDK9 inhibitors are currently in Phase I development such as Alvocidib, BAY 1143572, and TG02, and these may help to mitigate therapeutic resistance in VH patients [45].

In this investigation, only a limited number of DDR proteins for which there are molecules in development were able to be analyzed. Future analyses may include additional proteins and activation states that are known to participate in DDR signaling, for example CDK7 and many other proteins that were not assessed here. Our study is also restricted by the exclusive examination of population averages, rather than by combining reports of individual cell differences. There may be important subsets of cells with variable DDR expression that remained undetected by our analysis, but that could otherwise be revealed through techniques such as CyTOF that quantify a range of cellular components in single cells. 

In summary, DDR and DDR-related protein expression identified distinct AML patient clusters that significantly differed by clinical outcomes including RD and OS. Global protein activation patterns also revealed favorable and unfavorable clusters of CC and VH patients, thus offering guidance for potential chemotherapeutic combinations and targeted protein therapies in these patients.

## 4. Methods

### 4.1. Sample Collection and Processing

PB and BM samples were collected at baseline from 810 newly diagnosed adult AML patients evaluated at the University of Texas MD Anderson Cancer Center (MDACC) between January 2012 and July 2020. Informed consent was obtained in accordance with the Declaration of Helsinki. Samples were processed and analyzed under an Investigational Review Board (IRB)-approved laboratory protocol (Lab 06-0565). The following methods for sample collection and processing, protein lysate, and array slide printing were performed for all 810 adult patient samples as well as ten cryopreserved normal bone marrow derived CD34+ samples from healthy subjects (normal control). AML samples from 500 pediatric patients participating in the COG AAML1031 phase III clinical trial were processed similarly, as previously described by Hoff et al. [10,46].

For the samples, whole cell AML blast lysates were generated from fresh peripheral blood (*n* = 273) and bone marrow (*n* = 545) patient samples. Leukemic cells were enriched by Ficoll separation to isolate a mononuclear cell fraction followed by CD3/19 magnetic T and B cell depletion. To prepare lysates, the cell concentration was normalized to a concentration of 10,000 cells µL^−1^ and 10 million leukemia blast-enriched cells were suspended in 500 µL PBS, lysed in 500 µL of boiling hot protein lysis buffer (Tris buffered saline pH 7.4, 10% SDS, 2% beta-mercaptoethanol).

### 4.2. RPPA Methodology

Proteomic profiling of the patient samples was performed by RPPA as previously described [47,48,49]. Briefly, cell lysates were diluted in five serial dilutions in 96-well plates and transferred into 384-well plates. Plates were loaded into the Aushon 2470 arrayer and lysate material was printed onto nitrocellulose-coated glass slides with a single touch per dot. The five serial dilutions gave printed dots with approximately 85, 42, 21, 11, and 5 cell equivalents of protein, respectively.

Slides were probed with strictly validated primary antibodies, 412 in the adult RPPA and 296 in the pediatric RPPA, followed by a secondary antibody for signal amplification. Stained slides were scanned using the InnoScan 710 InfraRed microarray scanner and analyzed using Microvigene software version 3.4 to produce quantitative protein expression data. We used the nomenclature system previously described whereby post-translation modifications are denoted by a period that follows the protein name, then the type of post-translation modification, ‘p’ for phosphorylation, ‘cle’ for cleaved, and ‘Me’ for methylation, followed by the letter abbreviation for the affected amino acid and finally its sequence position [50].

### 4.3. Selected DDR and DDR-Related Proteins

Among the antibodies assessed on the adult RPPA, 21 detected total or PTM forms of DDR proteins: ‘BABAM1.pS29′, ‘BAP1′, ‘CHEK1′, ‘CHEK1.pS345′, ‘CHEK2′, ‘CHEK2.pT68′, ‘DDB1′, ‘MSH2′, ‘MSH6′, ‘PDCD1′, ‘RAD50′, ‘RAD51′, ‘RPA32′, ‘RPA32.pS4_8′, ‘SIRT6′, ‘SSBP2′, ‘TP53BP1′, ‘VCP’, ‘XPA’, ‘XPF also known as ERCC4′, and ‘XRCC1′. Satellite proteins reflective of the emerging crosstalk between DDR and other cellular signaling pathways were also analyzed in this report: ‘AURKA’, ‘AURORA_A_B_C.pT288_232_198′, ‘CDC25C’, ‘CDK1_2_3.pT14′, ‘CDK1′, ‘CDK2′, ‘CDK9′, ‘CDKN1A’, ‘CDKN1B.pS10′, ‘CDKN1B.pT198′, ‘CDKN1B’, ‘CDKN2A’, ‘CDT1′, ‘H2AX.pS139′, ‘PCNA’, and ‘TP53′ [29].

The pediatric RPPA assessed 21 DDR proteins that included 14 proteins also on the adult RPPA array (‘CHEK1′, ‘CHEK1.pS345′, ‘CHEK2′, ‘CHEK2.pT68′, ‘MSH2′, ‘MSH6′, ‘RAD50′, ‘RAD51′, ‘RPA32′, ‘RPA32.pS4_8′, ‘SSBP2′, ‘VCP’, ‘XPA’, and ‘XRCC1′) and 7 proteins not on the adult array (‘ATM’, ‘ATM.pS1981′, ‘BRCA2′, ‘CHEK1.pS296′, ‘ERCC1′, ‘ERCC5′, and ‘SIRT1′). COG specimens were collected to address a protocol-specified aim to examine for changes in protein expression patterns using RPPA.

### 4.4. Proteomic Analysis

Initial computational quality control steps were performed as previously described to ensure proper slide alignment, background noise control, and sample loading control [50]. The SuperCurve algorithm was used to generate a single expression value from the five serial dilutions [51]. Protein expression levels in AML patient samples were normalized relative to the median expression of 10 bone marrow-derived CD34+ samples from healthy subjects set to zero. Survival curves were generated using the Kaplan–Meier method and survival data was analyzed using a multivariate Cox regression model [52,53]. Standard analyses examined sextiles, terciles, and medians, where Bonferroni corrections or the Benjamini–Hochberg procedure were applied in cases of multiple statistical comparisons. Protein expression clusters were identified using unbiased hierarchical clustering. Correlations between protein clusters and clinical features were assessed using Fisher’s exact test for categorical variables and the Kruskal–Wallis test for continuous variables. Multivariate analysis for OS and RD was performed by Cox proportional hazards model. Median protein expression within clusters was compared to median expression in the normal controls to identify significant expression trends with the Wilcoxon test. All bioinformatics and analyses were performed in RStudio (version 1.3.1093) with R (version 4.1.2). Statistical analysis was not performed by COG.

## Figures and Tables

**Figure 1 ijms-24-05898-f001:**
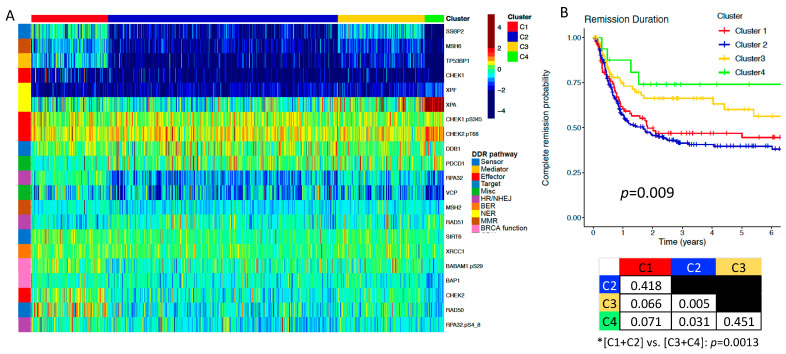
DDR proteins reveal four distinct clusters of adult AML patients with significantly different remission durations. (**A**) The core set of 21 DNA damage repair (DDR) proteins used to stratify AML patients into four separate clusters with significantly different remission durations (RD) are delineated as rows on the right side of the heatmap. The top of the heatmap indicates the cluster assignment (C1−C4) for each patient, while the heatmap itself displays levels of DDR protein expression in all patients. Proteins downregulated in expression relative to normal CD34+ cells are colored in cooler colors approaching deep blue, while upregulated proteins are colored in warmer colors approaching burgundy, as indicated by the color scale located at the top right of the heatmap. (**B**) The color of the remission duration curve for each cluster corresponds to the color band at the top of the heatmap (*n* = 398). Below the RD curves is a table displaying statistical comparisons and associated *p*−values between each individual cluster as well as [C1+C2] vs. [C3+C4].

**Figure 2 ijms-24-05898-f002:**
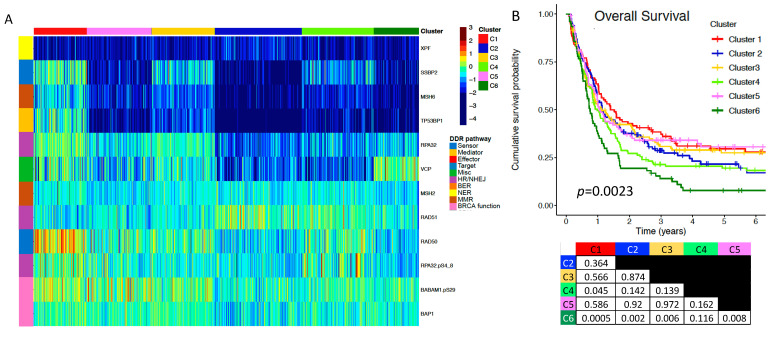
A subset of DDR proteins is prognostic for overall survival. (**A**) A subset of 12 DDR proteins (shown on the right side of the heatmap) from the original set of 21 reveals 6 AML patient clusters (labeled at the top of the heatmap) with distinct protein expression patterns. The heatmap itself displays the expression levels of this subset of DDR proteins in all patients. Proteins downregulated in expression relative to normal CD34+ cells are colored in cooler colors approaching deep blue, while upregulated proteins are colored in warmer colors approaching burgundy. (**B**) Maintaining the same cluster color designation, the overall survival (OS) for these 6 clusters is shown in the cumulative survival probability diagram (*n* = 684). Below the OS curves is a table displaying statistical comparison *p*-values between all combinations of cluster pairs.

**Figure 3 ijms-24-05898-f003:**
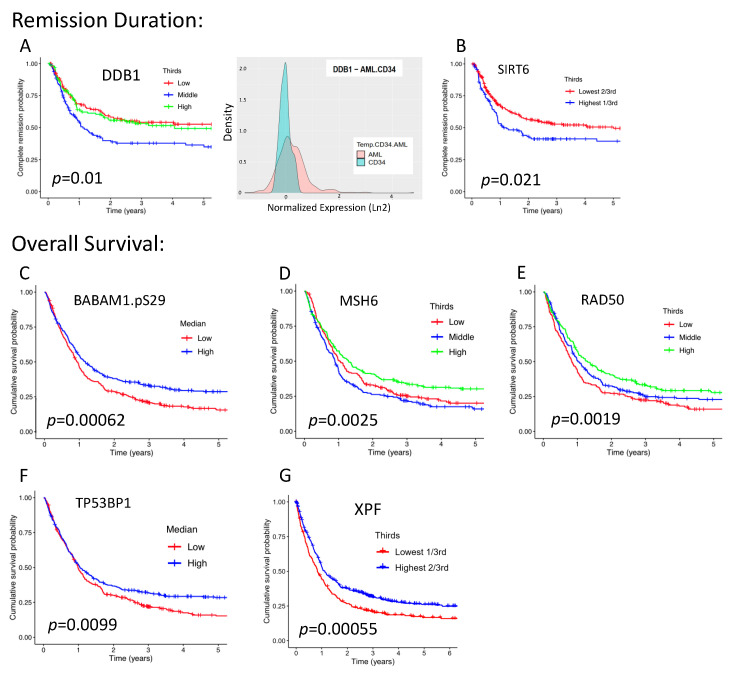
7 DDR proteins are individually prognostic for remission duration and/or overall survival. Each of the 21 DDR proteins was analyzed at multiple expression level splits to determine if any were significantly associated with clinical outcomes. Of the 21, only 7 were individually prognostic, and only the most significant splits for each protein were chosen for display. RD curves (*n* = 398) significantly stratified by (**A**) tercile of DDB1 expression and (**B**) lowest two-thirds and highest one-third of SIRT6 expression. The density graph insert to the right of (**A**) shows that AML cells display a much broader distribution of DDB1 expression than do normal CD34+ cells, whereby a significant proportion of AML cells are characterized by DDB1 overexpression. OS curves (*n* = 684) stratified by (**C**) median of BABAM1.pS29 expression, (**D**) tercile of MSH6 expression, (**E**) tercile of RAD50 expression, (**F**) median of TP53BP1 expression, and (**G**) lowest one-third and highest two-thirds of XPF expression.

**Figure 4 ijms-24-05898-f004:**
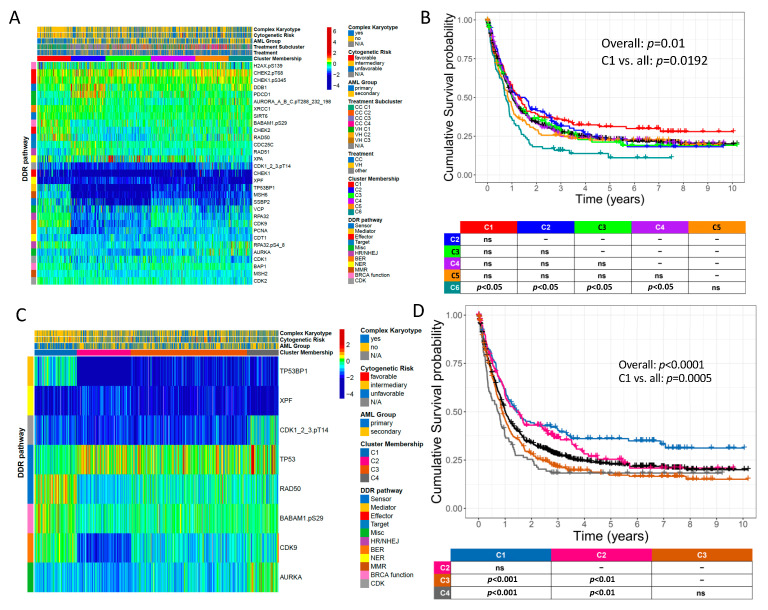
Novel DDR-related protein groupings are prognostic for overall survival. The core set of 21 DDR proteins was combined with 10 additional DDR-related proteins that have been previously cited in the literature to define an expanded array of DDR+DDR-related proteins (ExpDDR). (**A**) These 31 proteins are delineated as rows in the heatmap (right side), where each protein also has an annotation for its relevant DDR pathway (left side). The top of the heatmap depicts 6 unique clusters of patients with distinct proteomic profiles. Annotations including cytogenetic risk group and complex karyotype are also included at the top of the heatmap. The heatmap itself displays the expression levels of these DDR-related proteins in all patients. Proteins downregulated in expression relative to normal CD34+ cells are colored in cooler colors approaching deep blue, while upregulated proteins are colored in warmer colors approaching burgundy. (**B**) The associated Kaplan–Meier OS curves (*n* = 810) for each of the 6 identified patient clusters. Pairwise comparisons between the clusters are shown beneath the cumulative survival probability curves. (**C**) A subset of 8 maximally prognostic ExpDDR proteins identified four unique patient clusters with (**D**) associated survival curves (*n* = 810). Pairwise cluster comparisons are depicted beneath the OS curves. ns = not significant.

**Figure 5 ijms-24-05898-f005:**
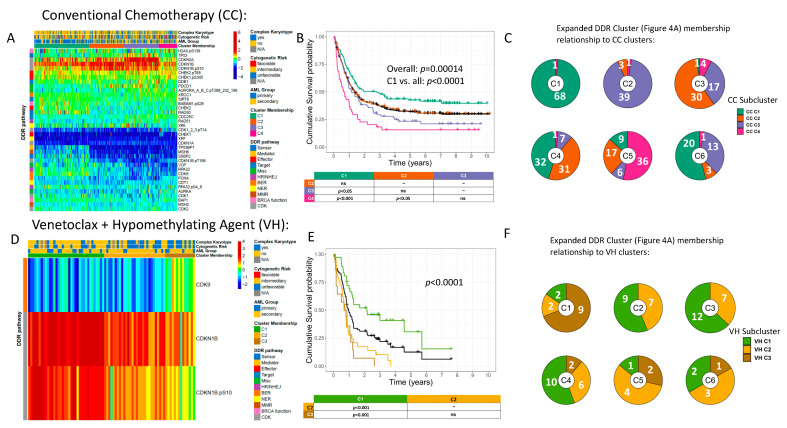
DDR and DDR-related protein groupings are prognostic for overall survival in CC and VH patients. The expanded set of DDR and DDR-related proteins (ExpDDR) was combined with additional DDR-related proteins [29] to constitute a maximally expanded array of relevant proteins involved in DDR. (**A**) Among those patients treated with conventional chemotherapy (CC), this maximally expanded protein grouping identified four patient clusters with distinct DDR expression patterns, as shown in the heatmap. The heatmap itself displays the expression levels of these DDR-related proteins in all patients. Proteins downregulated in expression relative to normal CD34+ cells are colored in cooler colors approaching deep blue, while upregulated proteins are colored in warmer colors approaching burgundy. (**B**) Associated OS curves (*n* = 340) for CC patients stratified by the same four clusters identified in the heatmap. (**C**) Pie charts show the proportion of these four clusters represented in each of the six clusters identified by the ExpDDR grouping in Figure 4A. (**D**) A subgroup of three proteins from the maximally expanded set of DDR and DDR-related proteins was prognostic for OS among patients treated with venetoclax and a hypomethylating agent. The heatmap is derived from three DDR-related proteins, CDK9, CDKN1B, and CDKN1B.pS10, and identifies three clusters of patients. (**E**) Associated OS for these three clusters (*n* = 79). (**F**) Expanded DDR Cluster (Figure 4A) membership relationship to VH clusters. When analyzed in conjunction with panel (**C**), the pie charts collectively identify which of the six ExpDDR clusters contain noticeable differences in outcome if patients were treated with CC vs. VH. ns = not significant.

**Figure 6 ijms-24-05898-f006:**
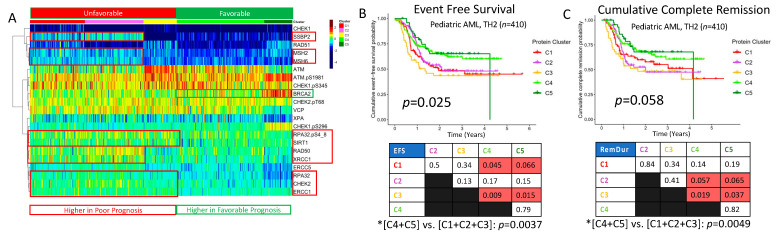
DDR proteins reveal five pediatric clusters with significantly different prognoses. An alternate set of 21 DDR proteins from the pediatric RPPA array was used to identify five clusters of pediatric patients with variable clinical outcomes. (**A**) Heatmap derived from 21 DDR proteins (*n* = 410) for five pediatric AML clusters. C1 (red), C2 (lavender), and C3 (yellow) represent unfavorable prognosis clusters while C4 (lime) and C5 (forest green) represent favorable prognosis clusters. Proteins downregulated in expression relative to normal CD34+ cells are colored in cooler colors approaching deep blue, while upregulated proteins are colored in warmer colors approaching burgundy. (**B**) Associated event-free survival (EFS), and (**C**) associated RD. Pairwise comparisons between clusters for both EFS and RD are shown beneath the cumulative probability curves in (**B**) and (**C**), respectively.

**Figure 7 ijms-24-05898-f007:**
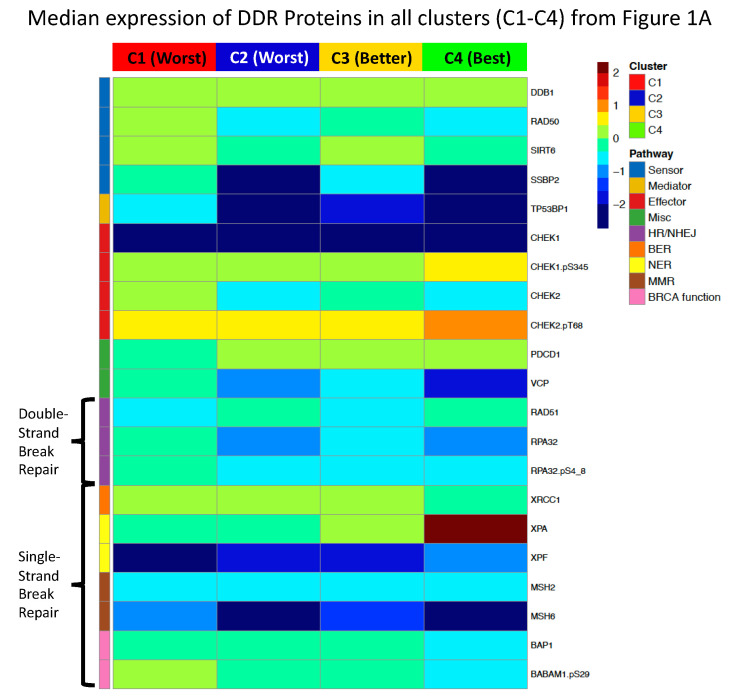
Unique global DDR patterns in patient clusters from Figure 1A. The heatmap displays the median expression of the 21 core DDR proteins for all four clusters from Figure 1A. Proteins downregulated in expression relative to normal CD34+ cells are colored in cooler colors approaching deep blue, while upregulated proteins are colored in warmer colors approaching burgundy. Notations for each protein’s functional role in DDR are displayed on the left side of the figure.

**Table 1 ijms-24-05898-t001:** DDR protein expression and activation are abnormal in adult and pediatric AML.

Adult Patients						
		Expression in AML Patents Relative to Normal CD34+ Control Range				
**Protein**	**AML Median**	**Below**	**Within**	**Above**	**% Above or Below**	**CD34 Range**
**BAP1**	**−0.1801**	**26%**	67%	7%	**33%**	[−0.4494, 0.6957]
**CHEK1**	**−0.3171**	**43%**	54%	4%	**46%**	[−0.3685, 0.3009]
**CHEK1.pS345**	**−2.4923**	**93%**	7%	0%	**93%**	[−1.3991, 0.9762]
**CHEK2**	**0.3325**	1%	54%	45%	**46%**	[−0.2368, 0.3813]
**CHEK2.pT68**	**−0.2762**	**34%**	57%	9%	**43%**	[−0.4641, 0.4964]
**DDB1**	**0.5638**	1%	72%	27%	**28%**	[−0.5936, 0.7839]
**MSH2**	**0.1621**	11%	51%	37%	**49%**	[−0.3157, 0.3599]
**MSH6**	**−0.5742**	**91%**	8%	0%	**92%**	[−0.1465, 1.6748]
**PDCD1**	**−1.7839**	**90%**	10%	0%	**90%**	[−0.7149, 0.718]
**RAD50**	**−0.0009**	5%	**91%**	4%	9%	[−0.616, 1.3787]
**RAD51**	**−0.3321**	**36%**	55%	9%	**45%**	[−0.5228, 0.7557]
**RPA32**	**−0.3817**	**57%**	32%	10%	**68%**	[−0.2974, 0.256]
**RPA32.pS4_8**	**−0.6294**	**54%**	41%	5%	**59%**	[−0.5541, 0.2611]
**SIRT6**	**−0.4622**	**64%**	25%	12%	**75%**	[−0.3254, 0.1799]
**SSBP2**	**−0.0022**	4%	84%	12%	16%	[−0.5078, 0.3751]
**TP53BP1**	**−1.4872**	**85%**	12%	3%	**88%**	[−0.3191, 0.2933]
**VCP**	**−2.0915**	**70%**	28%	2%	**72%**	[−1.5716, 0.1293]
**XPA**	**−0.7430**	17%	80%	2%	20%	[−1.5616, 0.6984]
**XPF**	**−0.1549**	**52%**	23%	25%	**77%**	[−0.1174, 0.3724]
**XRCC1**	**−1.9504**	**100%**	0%	0%	**100%**	[−0.5679, 0.5229]
**Pediatric Patients**						
		Expression in AML Patents Relative to Normal CD34+ Control Range				
**Protein**	**AML Median**	**Below**	**Within**	**Above**	**% Above or Below**	**CD34 Range**
**ATM**	**0.4680**	6%	58%	**35%**	**41%**	[−0.6544, 0.7334]
**ATM.pS1981**	**0.7960**	0%	44%	**56%**	**56%**	[−0.6671, 0.7286]
**BRCA2**	**0.3560**	1%	60%	**38%**	**39%**	[−0.5431, 0.4642]
**CHEK1**	**−2.8780**	**98%**	2%	0%	**98%**	[−1.1618, 1.1895]
**CHEK1.pS296**	**−0.1890**	0%	**97%**	3%	3%	[−1.5256, 0.739]
**CHEK1.pS345**	**0.6490**	1%	60%	**39%**	**40%**	[−0.4601, 0.7597]
**CHEK2**	**−0.2030**	21%	66%	13%	**34%**	[−0.5441, 0.2349]
**CHEK2.pT68**	**0.4500**	1%	18%	**81%**	**82%**	[−0.4548, 0.2165]
**ERCC1**	**−0.2050**	**45.0%**	47%	8%	**53%**	[−0.2362, 0.2293]
**ERCC5**	**−0.4420**	**48.0%**	49%	3%	**51%**	[−0.4588, 0.2613]
**MSH2**	**−0.9910**	**82.0%**	17%	0%	**82%**	[−0.5737, 0.3723]
**MSH6**	**−0.9510**	**45.0%**	54%	1%	**46%**	[−0.994, 0.4958]
**RAD50**	**0.0240**	15%	74%	11%	**26%**	[−0.4611, 0.646]
**RAD51**	**−1.6750**	**83.0%**	17%	0%	**83%**	[−0.9492, 0.638]
**RPA32**	**−0.3810**	6%	**93%**	1%	7%	[−1.0502, 0.3729]
**RPA32.pS4_8**	**0.0950**	9%	67%	24%	**33%**	[−0.3958, 0.3992]
**SIRT1**	**−0.0310**	9%	88%	3%	12%	[−0.6172, 0.7012]
**SSBP2**	**−1.3520**	**78.0%**	22%	0%	**78%**	[−0.5313, 2.0961]
**VCP**	**0.4120**	1%	61%	**38%**	**39%**	[−0.9309, 0.5162]
**XPA**	**−0.5240**	**36.0%**	56%	7%	**43%**	[−0.693, 0.4706]
**XRCC1**	**−0.0730**	6%	77%	17%	23%	[−0.7662, 0.3449]
	**AML Median**	**Below**	**Within**	**Above**	**% Above or Below**	
**Color Legend**	**median > 0.5**	**>25% Below Normal**	**>25% within Normal**	**>25% Above Normal**	**>75%**	
	**−0.5 < median < 0.5**	**>50% Below Normal**	**>50% within Normal**		**>50%**	
	**median < −0.5**	**>90% Below Normal**	**>90% Within Normal**		**>25%**	
	**median < −2.0**					

**Table 2 ijms-24-05898-t002:** Adult AML demographics and clinical information corresponding to Figure 1 clusters.

Clinical Variable	Unit of Measure	Cluster 1	Cluster 2	Cluster 3	Cluster 4	Overall	*p*-Value
		(N = 151)	(N = 451)	(N = 171)	(N = 37)	(N = 810)	
Gender	Female	74 (49.0%)	171 (37.9%)	72 (42.1%)	15 (40.5%)	332 (41.0%)	0.209
Age (Years)	Median	63.9	67.9	66.4	65.1	66.6	**0.003**
	Min, Max	22.8, 85.9	18.7, 94.0	19.9, 91.8	31.7, 85.4	18.7, 94.0	
Age Subgroup	18-40	14.60%	8.60%	16.40%	8.10%	92 (11.4%)	**0.009**
	41-60	31.10%	20.80%	24.60%	32.40%	195 (24.1%)	
	>60	54.30%	70.50%	59.10%	59.50%	523 (64.6%)	
Secondary AML	Yes	31.10%	55.70%	46.20%	54.10%	397 (49.0%)	**<0.001**
BM Blast Percentage	Median	72	30	55	29	41	**<0.001**
	Min, Max	2.0, 96.0	1.0, 95.0	1.0, 97.0	10.0, 85.0	1.0, 97.0	
WBC	Median	26.5	3.05	5.35	1.1	4.5	**<0.001**
	Min, Max	0.7, 391	0.3, 318	0.3, 363	0.3, 15.4	0.3, 391	
Hyperleukocytosis	>100,000	12.10%	4.30%	4.40%	0.00%	42 (5.7%)	**0.006**
PB Blast Percentage	Median	66	5	31	2.5	16	**<0.001**
	Min, Max	0, 98.0	0, 92.0	0, 94.0	0, 25.0	0, 98.0	
PB Absolute Blast Count	Median	16,600	138	1460	27	644	**<0.001**
	Min, Max	0, 379,000	0, 134,000	0, 247,000	0, 975	0, 379,000	
Platelet Count	Median	32	44.5	41	37	40	**0.024**
	Min, Max	6.0, 487	0, 1070	2.0, 1160	11.0, 122	0, 1160	
Platelets < 50,000	Yes	70.90%	53.30%	56.60%	70.40%	433 (58.0%)	**0.004**
Albumin	Median	3.6	3.9	4	4.3	3.9	**<0.001**
	Min, Max	2.0, 36.0	2.1, 7.3	2.1, 5.2	2.9, 4.8	2.0, 36.0	
Total Bilirubin	Median	0.5	0.7	0.6	0.6	0.6	**0.009**
	Min, Max	0, 2.2	0, 8.1	0.2, 3.0	0.2, 1.1	0, 8.1	
LDH	Median	1020	610	640	315	668	**<0.001**
	Min, Max	142, 10,600	117, 30,900	122, 16,000	168, 1230	117, 30,900	
Cytogenetics Risk Group	Favorable	5.00%	5.80%	4.50%	0.00%	38 (5.2%)	0.065
	Intermediate	66.90%	50.00%	55.80%	50.00%	398 (54.4%)	
	Unfavorable	28.10%	44.20%	39.60%	50.00%	295 (40.4%)	
Complex Cytogenetics	Yes	21.60%	35.70%	26.60%	50.00%	231 (31.6%)	**0.004**
ASXL1 Mutated	Yes	10.00%	23.10%	18.30%	41.70%	101 (19.9%)	**0.02**
FLT3.ITD Mutated	Yes	35.20%	9.00%	22.40%	5.90%	108 (16.8%)	**<0.001**
IDH1/2 Mutated	Yes	36.80%	20.10%	21.30%	12.50%	151 (23.4%)	**0.003**
NPM1 Mutated	Yes	43.90%	11.50%	17.10%	5.90%	119 (18.7%)	**<0.001**
RUNX1 Mutated	Yes	13.30%	16.00%	26.20%	36.40%	91 (18.1%)	**0.047**
TP53 Mutated	Yes	16.50%	25.50%	19.90%	50.00%	153 (23.4%)	**0.008**
WT1 Mutated	Yes	13.40%	3.90%	7.70%	9.10%	33 (6.7%)	**0.027**
Response	Remission	64.20%	57.80%	52.00%	69.60%	398 (58.2%)	0.411
	PR/HI/SD	5.20%	9.30%	8.70%	4.50%	56 (8.2%)	0.841
	Resistant	25.40%	29.40%	33.30%	27.30%	201 (29.4%)	
	Early Death < 28 days	5.20%	3.40%	6.00%	0.00%	29 (4.2%)	
Relapse	Yes	50.00%	53.70%	33.30%	25.00%	190 (47.7%)	**0.011**
% in CR at 5 years	%	44.00%	40.00%	60.00%	74.00%	301 (46.0%)	**0.009**
Survival (Weeks)	Median	72.7	54.4	47	122.4	55.6	0.15
	95% CI	53.1, 106.4	50.1, 61.4	42.6, 68	42.4, not_met	51, 62.9	

Please note: All measurements at time of diagnosis. PR/HI/SD = Partial response/Hematologic improvement/Stable disease. No significant differences in race, performance status, prior malignancy, prior chemotherapy, prior XRT, CNS leukemia, hemoglobin, ferritin, or serum creatinine at diagnosis.

**Table 3 ijms-24-05898-t003:** Targeted therapy agents involved in Phase I trials for proteins affected by DDR mutations.

Trials	Type	Agent 1	Agent 2	Agent 3
Funded IITs	PARP inhibitor combination	PARPi (Talazoparib)	VEGFRi (Axitinib)	
		PARPi (Talazoparib)	METi (Crizotinib)	
		PARPi (Talazoparib)	CDK4/6 (Palbociclib)	
	ATR inhibitor combination	ATRi (M6620)	PD-L1i (Avelumab)	
		ATRi (M6620)	DNA-Pki (M3814)	PD-L1i (Avelumab)
	Upcoming studies	PARPi (Niraparib)	PD-1i (Dostarlimab)	
NCI CTEP Trials	PARP inhibitor combination	PARPi (STAR: Seq. Olaparib)	WEE1i (Adavosertib)	
		PI3Ki (COD: Copanlisib)	PARPi (Olaparib)	
		PI3Ki (COD: Copanlisib)	PARPi (Olaparib)	PD-L1i (Durvalumab)
		PARPi (Talazoparib)	BETi (ZEN-3694)	
Strategic Alliance Trials	ATR inhibitor monotherapy	ATRi (Repair) (RP-3500)		
	ATR inhibitor combination	ATRi (RP-3500)	PARPi (Talazoparib)	
		ATRi (BAY-1895344)	PARPi (Niraparib)	
		ATRi (RP-3500)	PARPi (Olaparib or Niraparib)	
		ATRi (RP-3500)	Chemo (Gemcitabine)	
	NEW DDR agents	PK-MYT-1i (RP-6306)		
	Upcoming studies	MRTX849 (KRAS G12Ci)	PARPi (Olaparib)	

## Data Availability

Contact the corresponding author for access to the data.

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
