# Peer review of "DNA Damage Response−Related Proteins Are Prognostic for Outcome in Both Adult and Pediatric Acute Myelogenous Leukemia Patients: Samples from Adults and from Children Enrolled in a Children’s Oncology Group Study"

_ijms, 2023, doi:10.3390/ijms24065898_

Round 1

Reviewer 1 Report

In this study, the authors analyzed 21 proteins (or protein PTMs) that they describe as “DDR” proteins (DNA damage and repair). They did the analysis for AML patient samples and then related the levels of each protein (or modified protein) to a variety of parameters, including remission duration, cytogenetic risk, overall survival, and survival in CC and VH patients (conventional chemotherapy and treatment with venetoclax in combination with an HMA, respectively). This is an impressive study rich with nuances, and there are certainly key results of value. The major concerns of this reviewer are a) the rather odd selection of proteins in the analysis panel, b) the sheer number of comparisons that in sum tend to weaken the take-home messages, and c) the lack of discussion of DNA repair pathways. Furthermore, there is a disconnect between knowledge of what the consequences are for differing expression levels and the implications of the findings of the authors.

The major concern is the premise for the entire study. Why were these proteins picked? Is it known that differences in the levels of these proteins actually affect DNA repair? Why are there checkpoint proteins mixed in with repair proteins? It is as if there is an underlying assumption that the levels of these proteins meaningfully impact DNA repair, which may or may not be the case. At a minimum, there should be some discussion on the selection process, some information on whether or not the levels of each protein affect DNA repair capacity, and some discussion regarding which genes are part of which DNA repair pathway.

Also, why are the proteins for pediatric patients different from those for adult patients?

In Figure 3, for some genes, it is “highest or lowest 1/3” and for others it is “high, medium, low” and others are just “high and low”. It seems that the statisticians are picking and choosing to get the results they want to see. All the analysis methods should be consistent in this figure, else the implication is that there is bias.

There does not seem to be any rhyme or reason to Table 3. Why would Msh2 be different from Msh6? Why isn’t XPA on the list? Why are the designations for CDKN1Bs different depending on the phospho site? If sensible interpretation is not possible, then it is best to omit the results.

Given the rather random assortment of targets, rather than pointed questions about the relevance of particular pathways, conjecture about which treatment should be used for what conditions seems too much of a far reach.

What would happen if any random 21 proteins were picked? Would the same sorts of results follow? What is the evidence that what is being seen has anything at all to do with DNA repair capacity?

While it would be too much to ask for the analyses to be redone, at a minimum, at least some of these key concerns should be addressed.

Details:

It is not clear why PDCD1 is included on the list. This protein acts as part of CD8 T cell functions.

Page 2 line 8 1 should read “were”

There are a number of places where the font size changes.

Page 10 line 220 should read “VH”.

Author Response

Response to Reviewer 1 

 We think the attached  WORD file with embedded images will be easy to follow, but we have copied that text here as well. 

[Comment] In this study, the authors analyzed 21 proteins (or protein PTMs) that they describe as “DDR” proteins (DNA damage and repair). They did the analysis for AML patient samples and then related the levels of each protein (or modified protein) to a variety of parameters, including remission duration, cytogenetic risk, overall survival, and survival in CC and VH patients (conventional chemotherapy and treatment with venetoclax in combination with an HMA, respectively). This is an impressive study rich with nuances, and there are certainly key results of value. 

Response: Thank you, we appreciate your understanding. 

[Comment] The major concerns of this reviewer are a) the rather odd selection of proteins in the analysis panel, b) the sheer number of comparisons that in sum tend to weaken the take-home messages, and c) the lack of discussion of DNA repair pathways. Furthermore, there is a disconnect between knowledge of what the consequences are for differing expression levels and the implications of the findings of the authors. The major concern is the premise for the entire study. Why were these proteins picked? Is it known that differences in the levels of these proteins actually affect DNA repair? 

Response: The main goal of our proteomic studies was to provide a broader degree of proteomic profiling of leukemia than had previously been achieved. Most prior studies used Mass Spectrometry-based profiling, which while technically comprehensive, is not feasible for patient-derived samples, due to excessive sample requirements, costs, and time considerations. We therefore turned to RPPA, which is sample sparing and relatively significantly cheaper. As an antibody-based assay, it is dependent on the availability of highly specific, validatable antibodies. Since starting these arrays in 2005, we have now validated about 550 antibodies. Target proteins were chosen with the goal of providing a broad coverage of the known pathways and functionalities of a cell, focusing on those known to be involved in carcinogenesis, including known key, post-translationally modified forms (phosphorylated, methylated, etc.), when these are known. Not all interesting targets have an antibody that passes muster. So, by design and necessity, this is a cross section of proteins of known interest, but not a comprehensive profiling of all proteins. Among this broad set of proteins, 21 were known to be directly involved in DDR and were thus used as the core set for this current analysis. Importantly, this manuscript is one of three interrelated papers that collectively examine recurrent expression patterns of DDR proteins in leukemia. Figure 1 (copied below as this reviewer likely does not yet have access to that manuscript) from our accompanying paper entitled “Reverse Phase Protein Array Profiling Identifies Recurrent Protein Expression Patterns of DNA Damage Related Proteins Across Acute and Chronic Leukemia” classifies our selected proteins by their relevant biological function within the context of DDR.  These studies only measure a level of expression relative to that of normal CD34+ cells, so we cannot determine what level of expression corresponds to what degree of DNA repair. These findings serve as a launching pad for in-vitro and in-vivo experiments using cell lines or human samples (PDX mice) to test hypotheses suggested by this data.  

[Comment] Why are there checkpoint proteins mixed in with repair proteins? It is as if there is an underlying assumption that the levels of these proteins meaningfully impact DNA repair, which may or may not be the case. 

[Comment] Details: It is not clear why PDCD1 is included on the list. This protein acts as part of CD8 T cell functions. 

Response: We are uncertain if the reviewer is referring to PDCD1, a so called “checkpoint” regulating protein (which was also the subject of a comment originally 7 down from here, but moved up here for convenience of responding), or to the cell cycle checkpoint regulating proteins. So, we will address both here.    

Expression of PDCD1 is modulated by DNA double-strand break repair pathway activation. PDCD1 is also known to be affected by alternative DDR pathways including DNA damage ATR/Chk1 checkpoint signaling. These clarifying points have been added to the introduction along with their accompanying citations. 

These references are copied below for convenience: 

  • Sato H, Niimi A, Yasuhara T, Permata TBM, Hagiwara Y, Isono M, et al. DNA double-strand break repair pathway regulates PD-L1 expression in cancer cells. Nat Commun. 2017;8(1):1751. 
  • Zhang J, Shih DJH, Lin SY. Role of DNA repair defects in predicting immunotherapy response. Biomark Res. 2020;8:23. 
  • Mouw KW, Konstantinopoulos PA. From checkpoint to checkpoint: DNA damage ATR/Chk1 checkpoint signalling elicits PD-L1 immune checkpoint activation. Br J Cancer. 2018;118(7):933-5. 
  • Kakoti S, Sato H, Laskar S, Yasuhara T, Shibata A. DNA Repair and Signaling in Immune-Related Cancer Therapy. Front Mol Biosci. 2020;7:205. 

For the Cell Cycle Checkpoint Regulating proteins: 

CDK protein expression directly impacts DDR as the induction of cell cycle arrest is necessary for DNA repair and CDK levels control cell cycle progression. An explanation (with additional references) for our inclusion of CDK proteins in the analysis has been added to the introduction section. 

These references are copied below for convenience: 

  • Yu DS, Cortez D. A role for CDK9-cyclin K in maintaining genome integrity. Cell Cycle. 2011;10(1):28-32. 
  • Pruitt SC, Freeland A, Rusiniak ME, Kunnev D, Cady GK. Cdkn1b overexpression in adult mice alters the balance between genome and tissue ageing. Nat Commun. 2013;4:2626. 
  • Liontos M, Velimezi G, Pateras IS, Angelopoulou R, Papavassiliou AG, Bartek J, et al. The roles of p27(Kip1) and DNA damage signalling in the chemotherapy-induced delayed cell cycle checkpoint. J Cell Mol Med. 2010;14(9):2264-7. 
  • Cuadrado M, Gutierrez-Martinez P, Swat A, Nebreda AR, Fernandez-Capetillo O. p27Kip1 stabilization is essential for the maintenance of cell cycle arrest in response to DNA damage. Cancer Res. 2009;69(22):8726-32. 
  • Bencivenga D, Tramontano A, Borgia A, Negri A, Caldarelli I, Oliva A, et al. P27Kip1 serine 10 phosphorylation determines its metabolism and interaction with cyclin-dependent kinases. Cell Cycle. 2014;13(23):3768-82. 
  • Sun C, Wang G, Wrighton KH, Lin H, Songyang Z, Feng XH, et al. Regulation of p27(Kip1) phosphorylation and G1 cell cycle progression by protein phosphatase PPM1G. Am J Cancer Res. 2016;6(10):2207-20. 
  • Schiappacassi M, Lovisa S, Lovat F, Fabris L, Colombatti A, Belletti B, et al. Role of T198 modification in the regulation of p27(Kip1) protein stability and function. PLoS One. 2011;6(3):e17673. 
  • Wang Y, Sharpless N, Chang S. p16(INK4a) protects against dysfunctional telomere-induced ATR-dependent DNA damage responses. J Clin Invest. 2013;123(10):4489-501. 
  • Potrony M, Haddad TS, Tell-Marti G, Gimenez-Xavier P, Leon C, Pevida M, et al. DNA Repair and Immune Response Pathways Are Deregulated in Melanocyte-Keratinocyte Co-cultures Derived From the Healthy Skin of Familial Melanoma Patients. Front Med (Lausanne). 2021;8:692341. 
  • Liu K, Zheng M, Lu R, Du J, Zhao Q, Li Z, et al. The role of CDC25C in cell cycle regulation and clinical cancer therapy: a systematic review. Cancer Cell Int. 2020;20:213. 
  • Le Gac G, Esteve PO, Ferec C, Pradhan S. DNA damage-induced down-regulation of human Cdc25C and Cdc2 is mediated by cooperation between p53 and maintenance DNA (cytosine-5) methyltransferase 1. J Biol Chem. 2006;281(34):24161-70. 
  • Kciuk M, Gielecinska A, Mujwar S, Mojzych M, Kontek R. Cyclin-dependent kinases in DNA damage response. Biochim Biophys Acta Rev Cancer. 2022;1877(3):188716. 
  • Mayya V, Rezual K, Wu L, Fong MB, Han DK. Absolute quantification of multisite phosphorylation by selective reaction monitoring mass spectrometry: determination of inhibitory phosphorylation status of cyclin-dependent kinases. Mol Cell Proteomics. 2006;5(6):1146-57. 
  • Ma HT, Poon RYC. Aurora kinases and DNA damage response. Mutat Res. 2020;821:111716. 

Additionally, RPPA correlation analysis revealed that many of these CDK proteins were highly correlated with those proteins known to function specifically in DDR.  

[Comment] At a minimum, there should be some discussion on the selection process, some information on whether or not the levels of each protein affect DNA repair capacity, and some discussion regarding which genes are part of which DNA repair pathway. 

Response: Thank you, we have provided greater detail for the selection process in the introduction. A limitation of this study is that it lacks functional assays that determine whether or not alterations in the levels of each protein directly affect DNA repair capacity. We have added  clarification regarding which genes are part of which DNA repair pathway, as seen in Figure 1 of our accompanying paper “Reverse Phase Protein Array Profiling Identifies Recurrent Protein Expression Patterns of DNA Damage Related Proteins Across Acute and Chronic Leukemia,” which has already been accepted for this special issue. We have added a call out to this external figure in the introduction: “[Hoff, et. al.]” 

[Comment] Also, why are the proteins for pediatric patients different from those for adult patients? 

Response: The pediatric and adult RPPA arrays were assembled at different times (Pediatric ~ 2018, Adult in 2021) and we had validated more antibodies by the time the adult array was done. Results for the pediatric array also guided which antibodies we chose to use on the adult array, with “uninformative” antibodies not selected. Therefore, some proteins will overlap between the two arrays while others will not. 

[Comment] In Figure 3, for some genes, it is “highest or lowest 1/3” and for others it is “high, medium, low” and others are just “high and low”. It seems that the statisticians are picking and choosing to get the results they want to see. All the analysis methods should be consistent in this figure, else the implication is that there is bias. 

Response: There is no way to tell a priori if certain splits in protein expression will reveal significantly different outcomes. Is the prognosis effect continuous across the entire range of expression? Is there a threshold effect? Do both high and low levels have prognostic impact relative to a “normal” range of expression? If one looks for a single type of effect (for instance dividing at the median), then one lacks the ability to identify different types of effects. Thus, we risk discarding important biological data for the sake of statistical simplicity. We approach this data agnostically and do not assume that there is a particular effect, at a certain threshold or direction. We therefore have a standard analysis that looks at sextiles, thirds, medians, high 1/3rd, and low 1/3rd, but we also penalize ourselves for multiple searching. We therefore retain the ability to see different types of effects but stiffen our statistical rigor to avoid cherry picking. We typically apply a Bonferroni correction (in this case p-corrected = 0.05/5 measurements = 0.01) or the Benjamini-Hochberg procedure). The reviewer will note that in all cases in which multiple levels were used to search for an effect (Fig. 3), all the p-values are below this threshold except for SIRT6 (p=0.02). In other circumstances, when multiple searches were not done (Fig. 1 B, 2 B, 3 B, D, 4 B, E) the commonly used p<0.05 cutoff was used. Having been statistically rigorous, we then chose to present the data that showed the most significant trend that was apparent. In general, for these proteins it doesn’t really matter what cut point we use as the “significant” proteins remained so, regardless of the cut point evaluated. For instance, for BABAM1.pS29 or XPF, the plots used in the submitted figure are for medians, but the tercile splits (Low, Middle, High) revealed significant outcome stratifications (respectively p=0.0018, p=0.0059). If the reviewer would like, we can expand the text to include the above discussion, but we don’t think that really adds to the story.  

[Comment] There does not seem to be any rhyme or reason to Table 3. Why would Msh2 be different from Msh6? Why isn’t XPA on the list? Why are the designations for CDKN1Bs different depending on the phospho site? If sensible interpretation is not possible, then it is best to omit the results. 

Response: Thank you for your suggestions. Table 3 has been omitted. Supplemental Tables S2 and S3 provide information that was previously summarized in Table 3, so these tables remain. DDR and DDR-related proteins, such as XPA, that were not significantly prognostic for overall survival (OS) in either Conventional Chemotherapy (CC) patients or Venetoclax + Hypomethylating Agent (VH) patients at any of the tested quantiles (medians, terciles, quartiles, quintiles, and sextiles), do not appear in these tables. To clarify by example: CDKN1B.pS10 and CDKN1B.pT198 are two different phosphorylation states of CDKN1B that are recognized by two different antibodies. Thus, these two proteins will exhibit different levels of expression and have different prognostic values. 

[Comment] Given the rather random assortment of targets, rather than pointed questions about the relevance of particular pathways, conjecture about which treatment should be used for what conditions seems too much of a far reach. 

Response: We thought this was a very useful suggestion, and to try to address this we have developed a new figure that shows the median expression of each core DDR protein within the clusters from the primary analysis (Fig. 1A) sorted by DDR function (sensor, mediator, effector, target, and type of repair HR/NHEJ, BER, NER, MMR, BRCA). This enables an easier understanding of what different aspects of DDR are being affected in each group. Additionally, we have extensively modified the discussion, deleting many of the more speculative potential applications of drugs and instead focusing on specific settings illustrating how DDR is being affected. 

[Comment] What would happen if any random 21 proteins were picked? Would the same sorts of results follow? What is the evidence that what is being seen has anything at all to do with DNA repair capacity? While it would be too much to ask for the analyses to be redone, at a minimum, at least some of these key concerns should be addressed. 

Response: From a prior painful experience, where a statistician misaligned the clinical and protein data in the dataset, thereby scrambling everything, which resulted in absolutely nothing being prognostic, we know that random selection or scrambling the data leads to nothing of interest. 

To prove this, as requested, we generated a random set of 21 proteins from the 411 available proteins on the adult RPPA dataset using the sample() function in R (Random Set 1: “random_proteins.pdf”). The clustering algorithm was applied and then we searched for a prognostic impact. Respectively, for 2 to 10 possible groups, the p-values for OS were: 0.1, 0.22, 0.29, 0.33, 0.42, 0.55, 0.58, 0.64, 0.73, and for remission duration the p-values were: 0.9, 0.12, 0.17, 0.27, 0.39, 0.5, 0.59, 0.63, and 0.73. Thus, none were statistically significant at the p <= 0.05 level, and this was without any penalty for multiple searches. 

To further corroborate that randomness leads to uninteresting results, we generated a second random set of 21 proteins, by first alphabetizing the 411 proteins and then selecting a random protein every 20 proteins (Random Set 2: “random2_proteins.pdf”). In this way, we avoided selecting multiple forms, e.g. total and phospho, of the same protein, and we avoided selecting similarly named proteins from the same pathway or protein functional group. Thus, this set was functionally random as well as alphabetically random. Respectively, for 2 to 10 possible groups, the p-values for OS were: 0.13, 0.21, 0.36, 0.43, 0.55, 0.47, 0.42, 0.41, 0.17, and for remission duration the p-values were: 0.73, 0.46, 0.63, 0.63, 0.64, 0.74, 0.25, 0.23, 0.3. Once again, none were statistically significant at the p <= 0.05 level, and this was without any penalty for multiple searches. 

In summary, a random assortment of antibodies will form some clustering on unbiased hierarchical clustering, but it lacks functional meaning and is prognostically irrelevant. We have shown the heatmap for 5 clusters and the KM plot for these same 5 clusters as an example below (Random Set 1). We have included this here as a response to the reviewer, but not added this to the manuscript. We can add it to the manuscript if the reviewer thinks it will be informative to the reader. 

[Comment] Page 2 line 8 1 should read “were” 

There are a number of places where the font size changes. 

Page 10 line 220 should read “VH”. 

Response: Thank you, these corrections have been made. 

Reviewer 2 Report

The authors have done an extensive study

The manuscript can be accepted following the minor revisions 

The introduction is way to less. This is a not too routine study, you need to introduce basics and make the general readers too understand what is the background. The novelty of the study is not that clearly explained.

The figure legends need to be elaborated. Mere mention is not sufficient for this kind of a study.

The citations on the whole.....are less. Please cite more literature. Discussio  lacks citations. Pls discuss based on published reports.

The structure of the manuscript...I think is not IJMS style. Results after introduction?

The language as such is fine. It's just that there are places where you are not clearly describing stuff. As in the case of figure caption . 

Conclusions are okay and abstract too. Could be more focussed. There's a lot of blah blah.....narrow down to highlight ur findings.

Author Response

Response to Reviewer 2 

[Comment] The authors have done an extensive study. The manuscript can be accepted following the minor revisions. The introduction is way to less. This is a not too routine study, you need to introduce basics and make the general readers to3o understand what is the background. The novelty of the study is not that clearly explained. 

Response: The introduction has been expanded to provide more robust reasoning for conducting the presented analysis. Additional background information, especially regarding acute myelogenous leukemia, has also been added. Moreover, the manuscript has been updated to emphasize the novelty of our study. We have added multiple paragraphs at the end of the introduction further detailing how our core selection of 21 proteins (Fig. 1A) and expanded selection of cell cycle checkpoint proteins (subsequent figures) directly relate to DNA damage repair.   

[Comment] The figure legends need to be elaborated. Mere mention is not sufficient for this kind of a study. 

Response: Thank you, all figure legends have been expanded to provide greater detail. 

[Comment] The citations on the whole.....are less. Please cite more literature. Discussio  lacks citations. Pls discuss based on published reports. 

Response: We have added 27 additional references to our manuscript. Many of these citations can be found in the introduction section, however multiple citations have also been added to the discussion.   

[Comment] The structure of the manuscript...I think is not IJMS style. Results after introduction? 

Response: We are also more used to the intro, methods, results, conclusion, order, but we have confirmed that the IJMS template provided online displays the identical order of sections that we have used in our manuscript. We are happy to reorder sections as the journal editors desire. 

[Comment] The language as such is fine. It's just that there are places where you are not clearly describing stuff. As in the case of figure caption.  

Response: Thank you, the language of the full manuscript has been reviewed for improvement. More detailed descriptions have been added, including for all figure captions. 

[Comment] Conclusions are okay and abstract too. Could be more focused. There's a lot of blah blah.....narrow down to highlight ur findings. 

Response: We thought this was a very useful suggestion, and to try to address this we have developed a new figure (Fig. 7) in the discussion that shows the median expression of each core DDR protein within the clusters from the primary analysis (Fig. 1A) sorted by DDR function (sensor, mediator, effector, target, and type of repair HR/NHEJ, BER, NER, MMR, BRCA). This enables an easier understanding of what different aspects of DDR are being affected in each group. Additionally, we have extensively modified the discussion, deleting many of the more speculative potential applications of drugs and instead focusing on specific settings illustrating how DDR is being affected. 

Round 2

Reviewer 1 Report

The revised version of this manuscript is greatly improved and all of this reviewer's major concerns have been addressed.